# Overexpression of *MxbHLH18* Increased Iron and High Salinity Stress Tolerance in *Arabidopsis thaliana*

**DOI:** 10.3390/ijms23148007

**Published:** 2022-07-20

**Authors:** Xiaoqi Liang, Yingmei Li, Anqi Yao, Wanda Liu, Tianyu Yang, Mengfei Zhao, Bingxiu Zhang, Deguo Han

**Affiliations:** 1Key Laboratory of Biology and Genetic Improvement of Horticultural Crops (Northeast Region), Ministry of Agriculture and Rural Affairs/National-Local Joint Engineering Research Center for Development and Utilization of Small Fruits in Cold Regions/College of Horticulture & Landscape Architecture, Northeast Agricultural University, Harbin 150030, China; a13989332297@163.com (X.L.); mei990304@163.com (Y.L.); a02200199@neau.edu.cn (A.Y.); yty12980@163.com (T.Y.); z1991689991@163.com (M.Z.); 2Horticulture Branch of Heilongjiang Academy of Agricultural Sciences, Harbin 150040, China; haaslwd@126.com

**Keywords:** *Malus xiaojinensis*, *MxbHLH18*, salt stress, Fe stress, gene transformation

## Abstract

In the life cycle of apple, it will suffer a variety of abiotic stresses, such as iron stress and salt stress. bHLH transcription factors (TFs) play an indispensable role in the response of plants to stress. In this study, a new *bHLH* gene named *MxbHLH18* was separated from *Malus xiaojinensis*. According to the results of subcellular localization, *MxbHLH18* was localized in the nucleus. Salt stress and iron stress affected the expression of *MxbHLH18* in *Malus xiaojinensis* seedlings to a large extent. Due to the introduction of *MxbHLH18*, the resistance of *Arabidopsis thaliana* to salt, high iron and low iron was significantly enhanced. Under the environmental conditions of high iron and low iron, the overexpression of *MxbHLH18* increased many physiological indexes of transgenic *Arabidopsis* compared to wild type (WT), such as root length, fresh weight and iron content. The high level expression of *MxbHLH18* in transformed *Arabidopsis thaliana* can not only increased the content of chlorophyll and proline, as well as increasing the activities of superoxide dismutase (SOD), peroxidase (POD) and catalase (CAT); it also reduced the content of malondialdehyde (MDA), which was more obvious under high salt conditions. In addition, the relative conductivity, H_2_O_2_ content and O^2−^ content in transgenic *Arabidopsis* decreased under salt stress. Meanwhile, *Mx**bHLH18* can also regulate the expression of downstream genes associated with salt stress (*AtCBF1/2/3*, *AtKIN**1* and *AtCOR15a/b*) and iron stress (*AtIRT1*, *AtFRO2*, *AtNAS2*, *ATACT2*, *AtZIF1* and *AtOPT3*). Therefore, *MxbHLH18* can actively promote the adaptability of plants to the growth environment of salt and low and/or iron.

## 1. Introduction

In the whole life cycle of plants, they are often affected by various environmental factors that are not conducive to growth, such as saline alkali conditions, drought, high temperature, iron deficiency, cold and lack of nutrition [1,2,3,4,5,6,7]. In order to survive under various adverse conditions and carry out normal life activities, plants will produce different adaptation mechanisms [8,9]. Among them, the role of TFs (transcription factors) is particularly critical [10]. Under abiotic stress, the expression level of TFs in plants will increase rapidly, transmitting signals to downstream genes and activating the expression of these genes, and producing a series of physiological and biochemical changes in plants [11,12,13,14]. bHLH, MYB, AP2/ERF, bZIP and other TFs are closely related to the response of plants to different stresses [15,16].

For plants, their growth and development are inseparable from mineral elements, such as Fe, Mn, Cu and Zn [17]. Among them, Fe participates in various life activities of plants, and is indispensable in many activities such as respiration, photosynthesis and chlorophyll synthesis of plants [18,19]. However, in most soil types, especially alkaline soil, the solubility of iron is very low, and the content of free iron is far from the content required for normal plant life activities [20]. Therefore, the problem of iron deficiency needs great attention.

In nature, bHLH TF is common in plants [21]; its domain contains about 60 amino acids, and the bHLH domain has two conserved regions with different functions [22]. One is located at the N-terminal of the bHLH domain and adjacent to the HLH motif, which is called the basic region. The basic region consists of about 15 amino acids, including 6 common amino acid residues [23]. This region is mainly responsible for the amino terminal essential DNA binding in plant life processes [24]. There are numerous members of the bHLH gene family in plants, and many of them have been identified. At present, 110, 167, 66 and 183 bHLH TFs have been identified in sweet potato [25], Arabidopsis [26], sweet cherry [27] and sunflower [28], respectively. Both biotic and abiotic stresses affect plant growth and development. When under biological stress, bHLH TFs can rapidly regulate the response of plants to biological stress. In the process of plant evolution, plant genes will undergo mutation. This can be roughly divided into two resistance mechanisms to reduce or prevent the impact of adverse external environment [29]. One of them is that plant hormones are rapidly activated by stimuli caused by the external environment. For example, these stimuli can activate the immune system. This expression mechanism can help plants show resistance to external pressure [30]. The other mechanism is plant infection with pathogens, leading to allergic reactions. In the affected part, gene mutation can limit the transfer of pathogens, thus protecting plants. A large number of studies have reported that the bHLH TFs family participates in the response of plants to stress, and found that many bHLH TFs are induced by stress treatments such as low iron, low temperature, drought, high salt. Abiotic stresses such as drought or salt stress can have harmful effects on plants, such as hindering plant growth, making leaves wilt, and even causing plant death in serious cases [31,32]. Studies have found that the *PkbHLH2* gene can make plants exhibit drought tolerance through the inducible expression of its own target genes [33]. The *VvbHLH1* gene may enhance the salt tolerance of plants by regulating the content of flavonoids [34].

*Malus xiaojinensis* is an important semi-dwarf apple germplasm resource, which is unique to China. Moreover, according to previous studies, *M. xiaojinensis* is an iron efficient apple genotype [35]. In the past decade, some molecular components related to high tolerance to iron deficiency in *M. xiaojinensis* have been separated and researched [36,37]. Physiological and molecular components related to iron acquisition and transport in *M. xiaojinensis* have been reported in most of these studies, such as *MxIRT1*, *MxMYB1*, *MxSAMS*, *MxNAS1*, *MxbHLH01*, *MxVHA-c*, *MxIRO2*, and *MxCS1-3* [38]. In this study, a *bHLH* gene named *MxbHLH18* was isolated from *M. xiaojinensis*. In the experiment, the expression levels of *MxbHLH18* in different organs were detected, and the effects of iron, cold and salt stress on the expression of *MxbHLH18* were also investigated. Due to the overexpression of *MxbHLH18* in *Arabidopsis*, the tolerance of transgenic plants to salt stress was significantly improved, the contents of chlorophyll and proline were also increased, as well as the activities of SOD, POD and CAT were increased, but the relative conductivity, MDA, H_2_O_2_ and O^2−^ contents were decreased. More importantly, the overexpression of *MxbHLH18* in *A. thaliana* improved the tolerance of transgenic plants to high and low iron stress, and its physiological indicators changed significantly. Overexpression of *Mx**bHLH18* also upregulated the expression of iron stress-related downstream key genes *AtIRT1*, *AtFRO2*, *AtNAS2*, *AtACT2*, *AtZIF1* and *AtOPT3* and salt stress-related downstream genes *AtIRT1*, *AtFRO2*, *AtNAS2*, *AtACT2*, *AtZIF1* and *AtOPT3*. This result was found for the first time.

## 2. Results

### 2.1. Sequence Analysis of MxbHLH18

Analysis from protparam (http://www.expasy.org/tools/protparam.html accessed on 5 June 2021) showed that the complete open reading frame (ORF) of *MxbHLH18* was 1025 bp, and the predicted MxbHLH18 protein contained 341 amino acids (Appendix A). The theoretical molecular mass of *MxbHLH18* was 37.975 kDa with a theoretical isoelectric point of 7.64 and the average hydrophilia coefficient was −0.328. There was a bHLH domain of MxbHLH18 protein (Figure 1A), indicating that it belonged to the bHLH family. 

### 2.2. Phylogenetic Analysis of MxbHLH18

In order to understand the homology between bHLH TFs in plants, MxbHLH18 was compared with bHLH proteins from 13 different plants by using Mega 6.0 (http://www.bio-soft.net/tree/MEGA.htm accessed on 13 June 2021). The phylogenetic tree showed that MxbHLH18, MdbHLH18 (*Malus domestica*, XP_028965545.1), PdbHLH25 (*Prunus dulcis*, XP_034228728.1), PmbHLH25 (*Prunus mume*, XP_008236173.1), PabHLH25 (*Prunus avium*, XP_021814463.1), RcbHLH25 (*Rosa chinensis*, XP_024167014.1), PtbHLH18 (*Populus trichocarpa*, XP_002300300.3), JrbHLH25 (*Juglans regia*, XP_018844450.1), SobHLH18 (*Syzygium oleosum*, XP_030472507.1), HbbHLH25 (*Hevea brasiliensis*, XP_021675440.1), QlbHLH25 (*Quercus lobata*, XP_030944154.1), CsbHLH25 (*Citrus sinensis*, XP_006486799.1), PvbHLH25 (*Pistacia vera*, XP_031276893.1), HubHLH18 (*Herrania umbratica*, XP_021300267.1), PpbHLH25 (*Prunus persica*, ONH92137.1), AtbHLH18 (*Arabidopsis thaliana*, NP_001324635.1) were grouped into another cluster. Among them, MxbHLH18 had the closest genetic relationship with MdbHLH18 (*Malus domestica*, XP_028965545.1) (Figure 1B).

### 2.3. MxbHLH18 Was Localized in the Nucleus

After staining with 4′,6-diamidino-2-phenylindole (DAPI) (Figure 2F), it was found that *MxbHLH18*-GFP fusion protein was labeled to the nucleus (Figure 2E), while the control GFP was localized in the cytoplasm (Figure 2B). Therefore, MxbHLH18 is a nuclear-localized protein.

### 2.4. The Expression Patterns of MxbHLH18 in M. xiaojinensis

Under the control condition, the expression level of *MxbHLH18* in the new leaves was the highest, followed by the roots, and very low in the mature leaves and stems (Figure 3A). Under current low iron content (4 μM Fe-EDTA, ––Fe), high iron content (160 μM Fe-EDTA, ++Fe), salt (200 mM NaCl) and cold (2 °C) stress treatments, it was found that the expression level of *MxbHLH18* in the new leaves of *M. xiaojinensis* showed a trend such that it increased rapidly at first, increased to the maximum and then began to decline. The time to reach the maximum value was 6 h, 9 h, 9 h and 3 h, respectively (Figure 3B). The expression level of *MxbHLH18* in roots was almost the same as that in leaves, and began to decline slightly after reaching the maximum at 3 h, 6 h, 6 h and 9 h, respectively (Figure 3C). It can be seen from the above results that the up-regulated expression of *MxbHLH18* in roots and new leaves of *M. xiaojinensis* seedlings can be induced by low temperature, low iron, high iron and salt treatment.

### 2.5. Overexpression of MxbHLH18 Contributed to Higher Salinity Tolerance in Transgenic A. thaliana

Three strains of T_3_ generation transgenic *Arabidopsis* (S1, S3 and S4) with *Mx**bHLH18* gene and WT and UL (unload line) grown in the medium to four leaf stage were transferred to nutrient soil, watered with salt solution (200 mM) for 7 days, and then their survival rates were recorded with 15 nutrient pots. It can be seen that under normal conditions, the growth of WT, UL and transformation lines (S1, S3 and S4) are generally the same. However, when treated with salt stress for 7 days (salt stress for 7 days), the appearance of WT and UL plants was significantly inferior to that of transgenic *Arabidopsis* (S1, S3 and S4) (Figure 4A).

Under control conditions, there was almost no difference in the survival rates of WT, UL, S1, S3 and S4. However, after 7 days of growth under salt stress, only 18.5% of WT and UL lines of *Arabidopsis* survived, while the survival rates of S1, S3 and S4 transgenic lines were 78.4%, 77.5% and 77.6%, respectively. The survival rate of WT and UL type plants was significantly lower than that of transgenic plants under high salt conditions (Figure 4B).

In order to further clarify the role of *MxbHLH18* in plant response to salt stress, the contents of a variety of related physiological indicators and the activities of SOD, CAT and POD were also determined. Under the control condition, the content and activity of these substances in all strains were significantly decreased. However, under the condition of high salt concentration, the activities of SOD, POD and CAT as well as the contents of proline and chlorophyll in transgenic lines (S1, S3 and S4) were increased due to the overexpression of *MxbHLH18*. On the contrary, the contents of MDA, H_2_O_2_, O^2−^ and relative conductivity were reduced. It can be seen from the above results that the resistance and tolerance of *Arabidopsis* to salt stress can be enhanced by the increase of *MxbHLH18* expression (Figure 5).

### 2.6. Overexpression of MbMYB108 Promotes the Expression of Salt Stress-Related Genes

Salt responsive bHLH TF regulates target gene transcription by binding promoter G-box/E-box or GCG box elements, and participates in the ABA signal transduction pathway. Members of the bHLH family can regulate by acting on different target genes. Therefore, in this experiment, the expression levels of several important genes downstream of bHLH TF under high salt concentration were analyzed (Figure 6). After 7 days of growth of all lines under salt stress, it was obvious that, compared with the control, although the expression levels of these 6 genes were up-regulated in all *Arabidopsis* lines, the expression levels in WT and UL lines were far lower than those in *MxbHLH18* overexpression transgenic lines (S1, S3, S4), indicating that *AtCBF1*, *AtCBF2* and *AtCBF3* were positively regulated by *MxbHLH18*; thus, activating the expression of key genes *AtKIN1*, *AtCOR15a* and *AtCOR15b* under salt stress enhances the adaptability and resistance of plants to salt.

### 2.7. Overexpression of MxbHLH18 Enhanced High and/or Low Fe Stress Tolerance in Transgenic A. thaliana

Some of the T_3_ generation *MxbHLH18* transgenic *Arabidopsis* (S1, S3 and S4), WT and UL lines were transferred to 1/2 MS medium with different iron content (4 μM, 100 μM, 400 μM Fe-EDTA) to observe their phenotypes. Two weeks later, all plants (WT, UL, S1, S3 and S4) grew well under the condition of normal iron content. However, in the low iron environment (4 μM), the etiolation of WT *A. thaliana* and UL *A. thaliana* was obvious, and the root system was shorter, while *MxbHLH18*-OE lines (S1, S3 and S4) appeared normal. Similarly, when the iron content was high (400 μM), the appearance of WT and UL lines was not as good as that of transgenic *Arabidopsis* (S1, S3 and S4), but the roots of all lines were not long (Figure 7).

In order to further understand the role of *MxbHLH18* in Fe stress, related physiological indicators such as root length, fresh weight, iron content, and activities of SOD, CAT and POD were also measured. Under control conditions, the above physiological indicators of WT *A. thaliana*, UL *A. thaliana* and transgenic lines (S1, S3 and S4) were not significantly different. However, under high or low iron stress, the taproot length, fresh weight and iron content of WT and UL *Arabidopsis* were not as high as those of transgenic *Arabidopsis* (S1, S3 and S4), while the increase of *MxbHLH18* expression resulted in the increase of other indicators such as SOD, POD and CAT activities, proline and chlorophyll contents, except for the decrease of MDA content. It can be seen from these results that *Arabidopsis* can better adapt to iron stress due to the overexpression of *MxbHLH18* (Figure 8).

### 2.8. Overexpression of MbMYB108 Promotes the Expression of High and/or Low Fe Stress-Related Genes

The molecular mechanism of iron absorption in plants is mainly through the joint action of various enzyme genes and transporter genes, so as to achieve the ability to resist iron stress. bHLH TF plays an important role in the regulation of plant iron nutrition. Therefore, the expression level changes of several important genes *AtIRT1*, *AtFRO2*, *AtNAS2*, *AtACT2*, *AtZIF1* and *AtOPT3* located downstream of *MxbHLH18* TF under low and/or high iron treatment were studied in this experiment. Compared with the control condition (100 μM), the expression levels of 6 downstream genes were up-regulated under low iron stress (4 μM); the up-regulation of 6 genes in all *Arabidopsis* (WT, UL, S1, S3 and S4) and the expression of downstream genes in transgenic plants was significantly higher than that in WT and UL, indicating that *MxbHLH18* TF had a positive regulatory effect on *AtIRT1*, *AtFRO2*, *AtNAS2*, *AtACT2*, *AtZIF1* and *AtOPT3*. However, under high iron stress (400 μM), the expression of these 6 downstream genes decreased, and the decline was more obvious in transgenic plants (S1, S3 and S4), which showed that *MxbHLH18* can promote the expression of downstream iron stress related genes under low iron stress, but inhibits the expression of these genes under high iron stress, so as to regulate the Fe^2+^ balance in plant cells and improve the ability of plants to resist iron stress (Figure 9).

## 3. Discussion

The bHLH TF family is one of the most widespread TFs in eukaryotes, with numerous members and diverse functions [39]. At present, the research on bHLH TFs in animals is relatively deep, and they are divided into 6 categories and 45 families. The research on bHLH TFs in plants is relatively shallow, so reference is made to the classification standards of animals. Since the discovery of the first bHLH TF in maize, researchers have used a variety of omics research techniques such as transcriptomics, proteomics, bioinformatics, and genetics to deeply mine the function of plant bHLH proteins. Studies have shown that it not only participates in plant growth and physiological processes, but also plays an important role in plant stress [40]. A large number of studies have reported that many bHLH TFs are induced by stress treatments such as low iron, low temperature, drought and high salt.

In *Arabidopsis*, bHLH is the second largest class of TFs. More than 160 genes have been predicted to encode bHLH, but only about 30% of the functions have been roughly determined [41]. Liu et al. showed that transgenic plants overexpressing *bHLH122* were more resistant to drought, NaCl and osmotic stress than wild-type (WT) plants [42]. In contrast, *bHLH122* loss-of-function mutants were more sensitive to NaCl and osmotic stress than WT plants. These results suggested that *bHLH122* positively regulates drought, NaCl and osmotic signaling. Compared with *Arabidopsis*, the research on bHLH protein in apple is less, but in recent years, the research on this aspect is also deepening. Mao et al. analyzed apple (*Malus domestica*) by whole genome sequencing technology [43], and identified 188 MdbHLH proteins that may be associated with drought and salt stress. The bHLH-related protein At-MYC2 (rd22BP1) and the MYB-related protein AtMYB2 can specifically bind to the MYC and MYB recognition sites in the promoter sequence of the *rd22BP1* gene, respectively, to regulate the expression of ABA-inducible genes (such as *rd22*), thereby improving the stress resistance of transgenic plants [44]. Studies have shown that the apple bHLH TF MdCIbHLH1 (cold-induced bHLH1) is significantly induced under cold stress. MdCIbHLH1 protein specifically binds to the MYC recognition sequence in the *AtCBF3* promoter, and overexpression of *MdCIbHLH1* enhances cold tolerance in transgenic *Arabidopsis*. Ectopic expression of *MdCIbHLH1* upregulates the expression of *CBFs* genes through the CBF (C-repeat-binding factor) pathway, enhancing the cold tolerance of transgenic *Arabidopsis*, tobacco and apple seedlings [45]. Thus, *MdCIbHLH1* plays a role in cold tolerance in a CBF-dependent manner in different plant species. *MdCIbHLH1* reflects its role in cold tolerance, and *MdCIbHLH1* may be a suitable gene to overcome low temperature stress and improve apple crop productivity under low temperature conditions [46]. Kim et al. found the first calcium-binding bHLH TF (*AtNIG1*) which was confirmed to be involved in the plant salt stress pathway. Analysis of *Arabidopsis* mutants knocking out *AtNIG1*-1 showed that mutant plants were more sensitive to salt stress [47]. Further analysis showed that the survival rate, fresh weight, chlorophyll content and protein content of *AtNIG1*-1 plants were reduced under salt stress, suggesting that *AtNIG1* plays a key role in plant salt stress signaling. Zhou et al. found that the expression levels of *DREB1A*/*CBF3*, *RD29A*, *COR15A* and *KIN1* related stress genes were up-regulated by overexpressing the *OrbHLH2* gene of wild rice (*Oryza rufipogon*), thereby significantly improving the salt tolerance of plants [48]. Iron homeostasis is very important for normal growth and development of plants. Plants regulate and maintain iron homeostasis at the transcriptional level through a series of complex regulatory processes. A large number of studies has been confirmed that many TFs participate in iron response. It was found that *Ntb**HLH1* in tobacco is a positive regulator of iron deficiency response gene, which can maintain the balance of iron ions in plant cells under iron deficiency conditions and improve the tolerance of plants to low iron stress [49]. Three TFs, IRO, FIT and PYE are members of bHLH TF family, which indicates that bHLH TF family proteins are indispensable in the regulation of iron homeostasis.

In this study, a new *bHLH* gene named *MxbHLH18* was isolated from *M. xiaojinensis*. Analysis from protparam (http://www.expasy.org/tools/protparam.html accessed on 15 July 2021) showed that the complete ORF of *MxbHLH18* is 1025 bp, and the predicted MxbHLH18 protein contained 341 amino acids. The theoretical molecular mass of *MxbHLH18* is 37.975 kDa with a theoretical isoelectric point of 7.64 and the average hydrophilia coefficient is −0.328. By analyzing its conserved domain, it can be seen that MxbHLH18 protein has a bHLH conserved domain (Figure 1A), which is the landmark feature of bHLH family, indicating that it belonged to the bHLH family.

According to the phylogenetic tree, the homology among 16 bHLH proteins was high. *MxbHLH18* belongs to the bHLH family. The constructed phylogenetic tree showed that MxbHLH18 and MdbHLH18 (*Malus domestica*, XP_028965545.1) had the highest homology (Figure 1B).

The localization of MxbHLH18 protein in the nucleus can be predicted by Wolf PSORT (https://wolfpsort.hgc.jp/ accessed on 15 July 2021) analysis. More importantly, as shown in Figure 2, from the results of subcellular localization, *MxbHLH18*, like other bHLH TFs, is also a protein located in the nucleus. This result is consistent with the prediction results of online websites.

Compared with mature leaves, the expression of *MxbHLH18* gene in different tissues of *M. xiaojinensis* was different, and the expression was higher in new leaves and roots (Figure 3). From this expression pattern, it can be concluded that the expression pattern of *MxbHLH18* gene is tissue-specific. *MxbHLH18* may play an important role in the response of *M. xiaojinensis* seedlings to stress. The results showed that cold, salt, high-iron and low-iron treatments all significantly affected the expression level of *MxbHLH18* in *M. xiaojinensis*, and the gene expression of *M. xiaojinensis* under four abiotic stresses changed with time. Under low iron stress, the expression of *MxbHLH18* in new leaves and roots reached its peak at 6 h and 3 h after treatment, respectively. Under high iron and salt stress, the expression of *MxbHLH18* in new leaves reached a peak at 9h, while the expression in roots also reached a peak at 6h. This indicator showed that the *MxbHLH18* gene plays a key role in the process of plants participating in the response to low iron, high iron and salt stress, laying a foundation for the next step in exploring the function of *MxbHLH18* TF and its role in stress. After low temperature treatment, the expression of *MxbHLH18* was also up-regulated, but the peak value was relatively low. Therefore, selected low iron stress, high iron stress and salt stress were selected to treat transgenic *Arabidopsis*, and then verified the role of *MxbHLH18* gene in plants under stress.

After salt stress treatment on WT, UL and *MxbHLH18* transgenic *Arabidopsis*, the phenotypic changes of all plants were observed, and the survival rate after stress treatment was counted and analyzed. The results showed that transgenic *Arabidopsis* had stronger resistance to salt stress than WT and UL (Figure 4). In order to further study the mechanism of *MxbHLH18* genes in the process of plant resistance to salt stress, the physiological and biochemical indexes of *Arabidopsis* were determined, and the genes related to salt stress located downstream of *MxbHLH18* were analyzed

SOD, CAT and POD are key protective enzymes in plants. Their activities can be used as indicators of plant tolerance to stress [50]. In high salt environments, chlorophyll will be destroyed, resulting in yellowing of plants [51,52,53]. Hence, chlorophyll content can reflect the degree of stress on plants. The content of MDA is closely related to the degree of peroxidation of plant cell membranes. The higher the content of MDA, the deeper the degree of cell membrane damage [54,55]. Therefore, the damage to plants in harmful environments can be reflected by the contents of chlorophyll, proline, malondialdehyde and antioxidant enzymes [9,56]. Overexpression of *MxbHLH18* enables transgenic plants to maintain good growth status under salt stress (Figure 4), and changes many physiological indicators; for example, SOD, POD and CAT activities, proline and chlorophyll content increased, while MDA, H_2_O_2_, O^2−^ content and relative conductivity decreased (Figure 5). The resistance of transgenic *Arabidopsis* to salt stress may be improved by these physiological indexes changed by *MxbHLH18*.

In order to understand the role of *MxbHLH18* introduced into *Arabidopsis* in iron stress, the phenotypic changes of all plants (WT, UL S1, S3 and S4) were observed and the survival rate was counted and analyzed after iron stress treatment. The results showed that overexpression of *MxbHLH18* in *Arabidopsis* greatly improved the adaptability of transgenic *Arabidopsis* to high and/or low iron stress. The appearance of all transgenic *Arabidopsis* (S1, S3 and S4) was better than that of WT and UL when treated with high iron and/or low iron stress. Overexpression of *MxbHLH18* enhanced the tolerance of transgenic *Arabidopsis* to iron stress (Figure 7), and increased root length, fresh weight, and iron content in *Arabidopsis*; it also resulted in increased SOD, POD, and CAT activities, the contents of amino acids and chlorophyll increased, and the content of MDA decreased under the treatment stress (Figure 8). *MxbHLH18* may increase iron tolerance by altering these physiological parameters in transgenic *Arabidopsis* under stress. In order to further analyze the role of genes in plant response to iron stress, the expression of downstream genes of *MxbHLH18* associated with iron stress was also analyzed.

TFs plays an extremely important role in the response of plants to stress, which will affect the expression of a variety of stress resistance genes. Studies have shown that when plants are subjected to osmotic stress, members of the bHLH family can act on different target genes. After 7 days of high salt stress, the expression levels of several bHLH TF downstream genes analyzed in this experiment were significantly higher in all plants than in the control, especially in the transgenic lines. The results showed that *AtCBF1*, *AtCBF2* and *AtCBF3* were positively regulated by *MxbHLH18*, thus activating the expression of key genes *AtKIN1*, *AtCOR15a* and *AtCOR15b* under salt stress (Figure 6).

The absorption and transportation of iron by plants can be regulated by bHLH TF through various mechanisms, so as to regulate iron nutrition in plants. Compared with the control, after low iron stress and high iron stress treatment, the expression levels of *AtIRT1*, *AtFRO2*, *AtNAS2*, *AtACT2*, *AtZIF1* and *AtOPT3* genes in all *Arabidopsis* (WT, UL, S1, S3 and S4) were changed; the expression levels of 6 downstream genes were up-regulated under low iron stress, and the up-regulation of 6 genes in transgenic *Arabidopsis* (S1, S3 and S4) with *MxbHLH18* gene was significantly higher than that of WT and UL, indicating that MxbHLH18 TF had a positive regulatory effect on *AtIRT1*, *AtFRO2*, *AtNAS2*, *AtACT2*, *AtZIF1* and *AtOPT3*. However under high iron stress, the expression of these 6 downstream genes decreased, and the decline was more obvious in transgenic plants (S1, S3 and S4), which showed that *MxbHLH18* can promote the expression of downstream iron stress related genes under low iron stress, but inhibit the expression of these genes under high iron stress, so as to regulate the Fe^2+^ balance in plant cells, thereby improving the ability of plants to resist iron stress (Figure 9).

To sum up, according to the above results and previous studies, a possible model has been derived, which describes how *MxbHLH18* plays a role under high salt, low and/or high iron stress (Figure 10). First, salt stress induces the expression of *MxbHLH18*, enabling it to bind to the CBF promoter. Three key genes in the regional CBF dependent pathway, *CBF1*, *CBF2* and *CBF3*, directly activate their expression by combining with the CRT/DRE cis-acting elements of the downstream salt stress response genes *AtKIN1*, *AtCOR15a* and *AtCOR15b*, thereby regulating the synthesis of ABA and ABA signal transduction; they jointly regulate the response of plants to salt stress through these two pathways. Secondly, the *MxbHLH18* gene is overexpressed due to the stimulation of low iron, bHLH protein and FIT protein form heterodimer, which induces the up-regulated expression of *AtFRO2* and *AtIRT1*. *AtFRO2* promotes the reduction of Fe^3+^ adsorbed on the root epidermis to Fe^2+^. *AtIRT1*, as a transporter gene of Fe^2+^, transports Fe^2+^ reduced by *AtFRO2* to root cells for use by plants. In addition, the expression of *AtNAS2* also increased, promoting the synthesis of MAs. Moreover, the expression of other iron stress key response genes *AtACT2*, *AtZIF1* and *AtOPT3* located downstream of *MxbHLH18* was also up-regulated. However, under high iron stress, the expression of these downstream genes was inhibited, reducing the absorption and transport of Fe^2+^ by plant cells, thereby maintaining the balance of Fe^2+^ in cells, indicating that *MxbHLH18* can regulate iron stress by regulating iron stress related genes.

## 4. Materials and Methods

### 4.1. Plant Material and Growth Conditions

The tissue culture seedlings of *M. xiaojinensis* were cultured in Murashige and Skoog (MS) medium, 1.2 mg/L indole butyric acid (IBA) was added to the medium, and the roots were produced after 45 days of culture. To achieve the purpose of further growth of seedlings, transfered them to Hoagland solution. The solution needs to be changed 3 times a week. The temperature of the incubator was set at 20 °C and the relative humidity was set at about 85%, and this state was maintained. When the seedlings had 6–7 mature leaves (fully developed), they were subjected to salt stress (200 mM NaCl concentration), low temperature stress (placed in a 2 °C incubator), low Fe stress (4 μM Fe-EDTA Hoagland solution) and high Fe stress (160 μM Fe-EDTA Hoagland solution). The control group was seedlings grown in Hogland solution (40 μM Fe-EDTA Hoagland solution) at 20 °C. After 0, 1, 3, 6, 9 and 12 h, sealed the leaves (including mature leaves and new leaves that are not fully expanded), roots and stems of all treated plants, and immediately put them into liquid nitrogen for freezing treatment, and then store them in a refrigerator at −80 °C to prepare for subsequent RNA extraction [57].

### 4.2. Isolation and qPCR Analysis of MxbHLH18

The total RNA of *M. xiaojinensis* under normal conditions and stress treatment were extracted by CTAB method. The extracted parts were root, stem, new leaf and mature leaf [58]. With TransGen’s TransScript^®^ First-Strand cDNA Synthesis Super Mix (TransGen, Beijing, China) synthesized the first strand of the cDNA of *M. xiaojinensis*. Using the first strand cDNA of *M. xiaojinensis* as a template, the complete sequence of *MxbHLH18* was successfully obtained by PCR. According to the homologous region of *MxbHLH18* (MDP0000256514), a pair of primers (*MxbHLH18*-F and *MxbHLH18*-R, Appendix A) were designed to amplify the full length of *MxbHLH18*. The DNA was obtained by thermal cycling, the process was as follows: 94 °C lasts for 2 min, 35 cycles were carried out, 94 °C continued for 30 s, then 53 °C lasted for 30 s, and 72 °C lasted for 30 s. The DNA fragments purified by gel were cloned into pEasy-T1 vector (TransGen) before sequencing (BGI, Beijing, China).

Previously, the qRT-PCR expression analysis method of *MxbHLH18* in *M. xiaojinensis* referred to Han et al. [59]. The qRT-PCR primers (*MxbHLH18*-qF and *MxbHLH18*-qR, Appendix A) were designed according to part of the sequence of *MxbHLH18*. The control group was the Begonia *Actin* gene (AB638619) which could be stably expressed under diverse conditions [60]. *Actin* primers (*Actin*-F and *Actin*-R, Appendix A) were designed in accordance with the sequences in GenBank database. The thermal cycle process was as follows: 40 cycles are carried out after 94 °C lasted for 5 min, 94 °C continued for 5 s, 58 °C lasted for 40 s, and 72 °C lasted for 15 s. Calculated the relative expression according to the 2^−∆∆CT^ method [61].

### 4.3. Sequence Analysis of MxbHLH18

DNAMAN6.0 software (Lynnon Biosoft, Quebec, QC, Canada) was used to analyze the amino acid sequence of MxbHLH18. The theoretical isoelectric point, relative molecular weight and total average hydrophilic coefficient of MxbHLH18 protein were all predicted according to ExPASY database (https://web.expasy.org/protparam/ accessed on 5 April 2022). On the basis of the similarity with the MxbHLH18 protein in the NCBI website (http://www.ncbi.nlm.nih.gov/blast/ accessed on 5 May 2022), the bHLH amino acid sequences of other species were selected from the blast results. Constructed homologous evolutionary tree with MEGA7.0 program (http://www.megasoftware.net/ accessed on 13 June 2021).

### 4.4. Subcellular Localization of the MxbHLH18 Protein

In order to construct the transient expression vector, the open reading frame (ORF) of *MxbHLH18* was cloned between *Xma*I and *Xbal*I sites of pSAT6-RFP-N1 vector. A modified red-shifted green fluorescent protein (GFP) existed in the *Xma*I-*Xbal*I sites of this vector. Transformation of *MxbHLH18*-GFP construct into onion (*Allium cepa*) epidermal cells by particle bombardment [62]. DAPI staining was regarded as a nuclear marker in the process of nuclear detection. The transient expression of *MxbHLH18*-GFP fusion protein was observed under confocal microscope (EVOS Floid) [38].

### 4.5. Generation of Transgenic A. thaliana Overexpressing MxbHLH18

The restriction endonuclease sites of *Xma*I and *Xbal*I were inserted into the ORF sequence of *MxbHLH18* by PCR to construct the overexpression vector for transforming *A.thaliana*. For constructing pCAMBIA2300-*MxbHLH18* overexpression vector, PCR products and pCAMBIA2300 were linked together by T_4_ DNA ligase, which were digested by *Xma*I and *Xbal*I. Then, under the transformation of LBA4404 mediated by Agrobacterium, the recombinant plasmid (pCAMBIA2300-*MxbHLH18*: *MxbHLH18* driven by CaMV 35S promoter) was successfully introduced into *Arabidopsis* (Columbia-0) [63]. The transgenic strains were screened in 1/2 MS medium containing 50 µg/mL kanamycin sulfate. WT and UL were used as controls, and the results of qPCR were analyzed to confirm the transgenic lines. The strains used in the following studies are all T_3_ generation transgenic plants.

### 4.6. Fe and Salt Stress Treatments to Transgenic A. thaliana

All plants were cultured on the medium for about 10 days until 4 true leaves were grown. 30 seedlings (WT, UL, S1, S3 and S4) of each strain were carefully transferred to the new MS medium, and added respectively 100 μM (normal level), 4 μM (low concentration) and 400 μM (high concentration) Fe to simulate normal conditions, low iron stress and high iron stress. After 14 days, observed their growth and development [64,65], and measured the root length (5 longer roots per row) and fresh weight.

As for salt stress, WT, UL and transgenic plants (S1, S3 and S4) of *A. thaliana* were planted in soil for two weeks. All strains were grown under controlled conditions and high salt environment (200 mM NaCl). Their survival rate after 7 days was recorded with 15 nutrient pots [66,67].

### 4.7. Determination of Relevant Physiological Indexes

Collected all materials of the above lines for measurement. The physiological indexes of all *Arabidopsis* strains (WT, S1, S3 and S4) were determined according to the following methods. Fresh weight and primary root length were measured according to the results of Han et al. [38]. The contents of iron and chlorophyll refer to the methods of Liu et al. [68] and Xu et al. [69] respectively. The activities of SOD, POD and CAT were determined according to the methods of Pan et al. [70], Ranieri et al. [71] and Zhang et al. [72]. The contents of proline and malondialdehyde were determined by the method of Jiang et al. [73]. Studied the method of Chen et al. [74] to measure the relative conductivity, the content of hydrogen peroxide was determined by spectrophotometry [75]. The measurement of superoxide anion content was determined by the method of Wang et al. [76].

### 4.8. Expression Analysis of MbMYB108 Downstream Genes

The mRNA of WT, UL and transgenic *Arabidopsis* (S1, S3 and S4) under low iron stress, high iron stress, salt stress and control conditions were extracted respectively, and the first strand cDNA which was used as a templatewas obtained by reverse transcription test. With *AtActin* as an internal reference, several downstream regulatory genes of bHLH transcription factor: salt stress response related genes (*AtKIN1*, *AtCOR15a*, *AtCBF1*, *AtCBF2*, *AtCBF3*, *AtCOR15b*) and iron stress response key genes (*AtIRT1*, *AtFRO2*, *AtNAS2*, *AtACT2*, *AtZIF1*, *AtOPT3*). The specific primers used are shown in Appendix A. The reaction system was the same as Section 4.2.

### 4.9. Statistical Analysis

One-way analysis of variance (ANOVA) was analyzed with SPSS Statistical 17.0 software (IBM China Co., Ltd., Beijing, China). Each index was measured five times, all tests were repeated three times and the standard error (±SE) was measured respectively. Statistical differences were referred to as significant * *p* ≤ 0.05, ** *p* ≤ 0.01.

## 5. Conclusions

In this study, a new *bHLH* gene which was identified as *MxbHLH18* was isolated from *M. xiaojinensis*. According to the results of subcellular localization, MxbHLH18 protein is localized in the nucleus. Moreover, the expression of *MxbHLH18* in *M. xiaojinensis* seedlings was greatly affected by salt, low iron and/or high iron environmental conditions. In addition, the adaptability and resistance of transgenic *Arabidopsis* to salt, low iron and/or high iron stress may be improved due to the overexpression of *MxbHLH18*. Moreover, overexpression of *MxbHLH18* will also regulate the expression of downstream genes and further improve the tolerance of plants to salt, low iron and/or high iron stress.

## Figures and Tables

**Figure 1 ijms-23-08007-f001:**
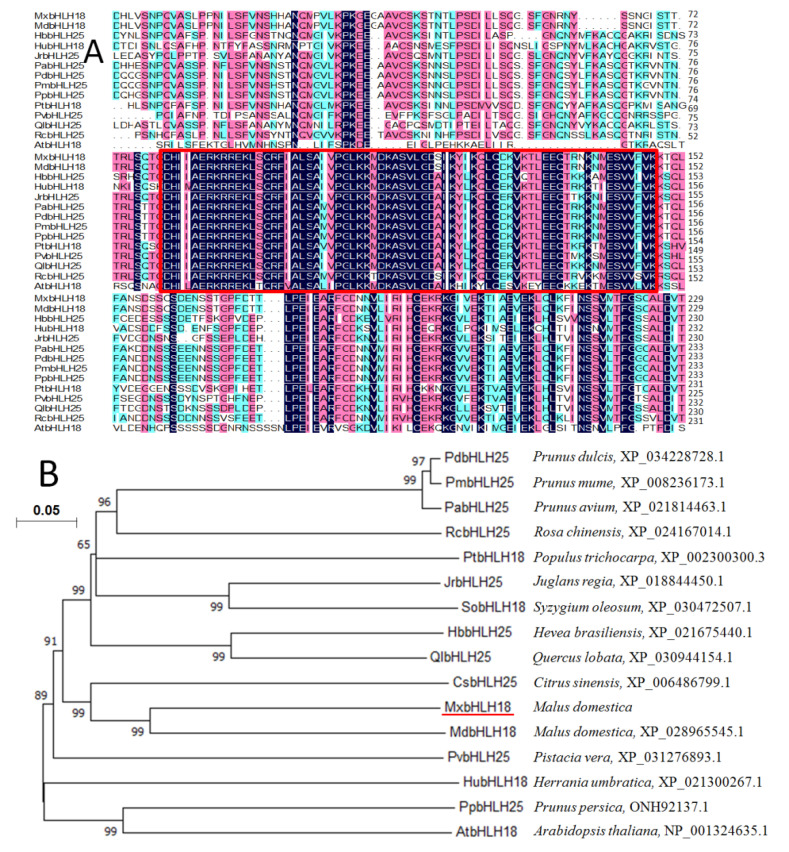
Comparison of amino acid sequences and phylogenetic relationship between *MxbHLH18* and bHLH from other species. (**A**) In the red box are the conserved domains of bHLH. (**B**) Phylogenetic tree analysis of bHLH proteins in MxbHLH18 (marked by red line) and other different plants. The accession numbers are as follows: MdbHLH18 (XP_028965545.1), PdbHLH25 (XP_034228728.1), PmbHLH25 (XP_008236173.1), PabHLH25 (XP_021814463.1), RcbHLH25 (XP_024167014.1), PtbHLH18 (XP_002300300.3), JrbHLH25 (XP_018844450.1), SobHLH18 (XP_030472507.1), HbbHLH25 (XP_021675440.1), QlbHLH25 (XP_030944154.1), CsbHLH25 (XP_006486799.1), PvbHLH25 (XP_031276893.1), HubHLH18 (XP_021300267.1), PpbHLH25 (ONH92137.1) and AtbHLH18 (NP_001324635.1).

**Figure 2 ijms-23-08007-f002:**
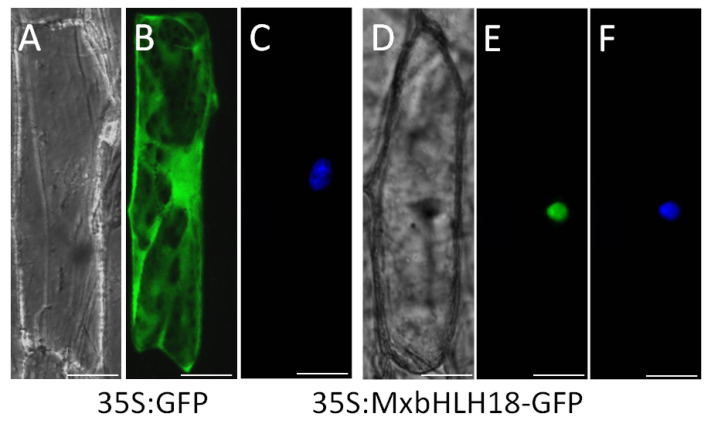
MxbHLH18 protein was localized in the nucleus. The 35S:GFP (control) and 35S:MxbHLH18-GFP fusion protein were expressed in onion epidermal cells. The expression results were observed under fluorescent microscope under strong light (**A**,**D**), dark field (**B**,**E**) and DAPI staining images (**C**,**F**). Scale represents 5 μM.

**Figure 3 ijms-23-08007-f003:**
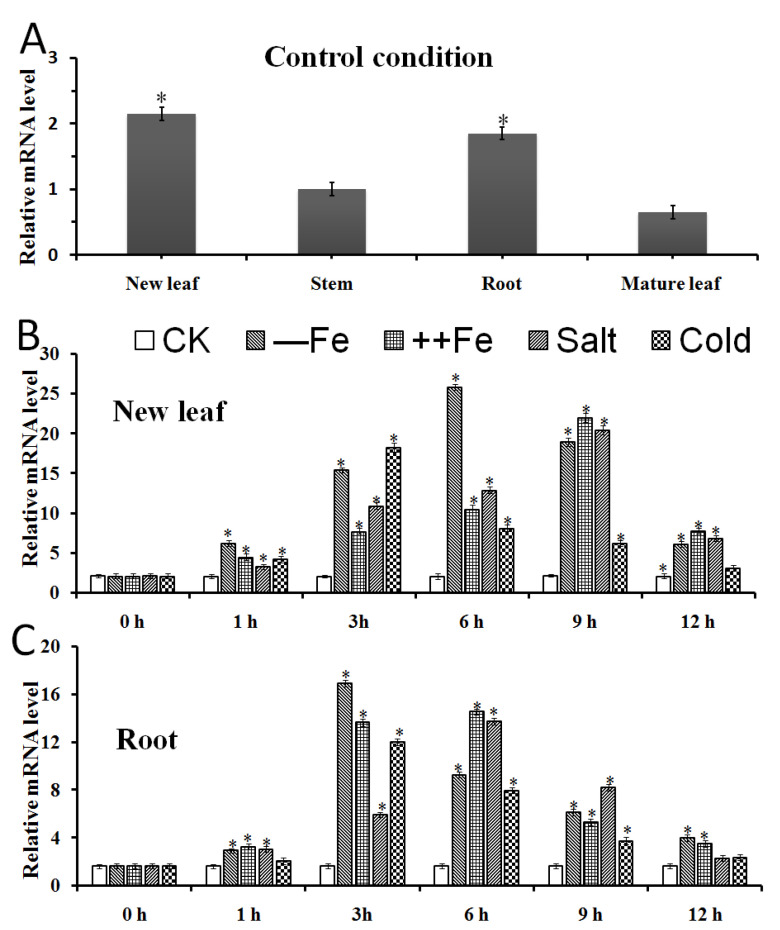
Expression patterns of *MxbHLH18* in different organs and quantitative real-time PCR results of responses to different stresses in *M. xiaojinensis*. (**A**) Under control conditions (40 μM Fe-EDTA), the expression level of *MxbHLH18* in roots, stems and leaves (new and mature leaves). The asterisk above the column represents a significant difference compared with the expression in the mature leaf (* *p* ≤ 0.05). (**B**,**C**) Under control condition (CK), low concentration iron (4 μM Fe-EDTA, ––Fe), high concentration iron (160 μM Fe-EDTA, ++Fe), low temperature (2 °C) and salt stress (200 mM NaCl), the expression level of *MxbHLH18* gene in new leaves and roots. The data represent the mean and standard error of three repetitions. The asterisk on the column indicates that the expression level is significantly different from that under the control condition (* *p* ≤ 0.05).

**Figure 4 ijms-23-08007-f004:**
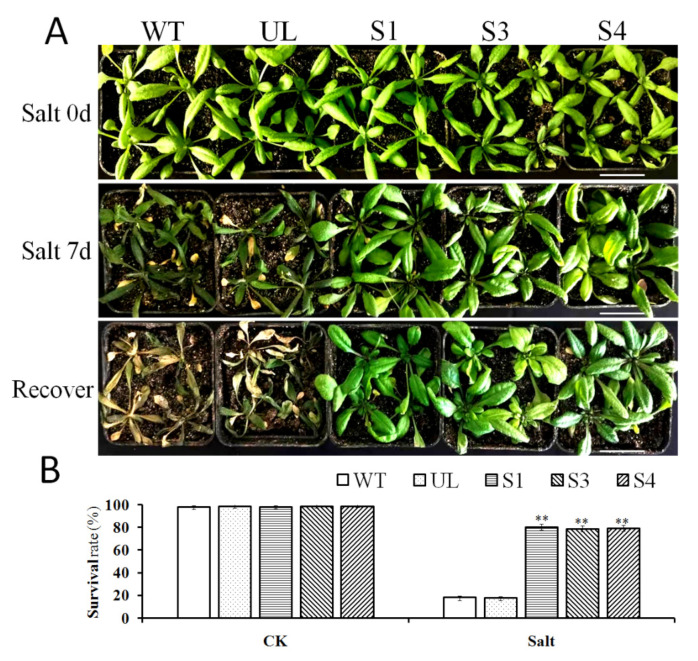
Overexpression of *MxbHLH18* can improve salt tolerance in *A. thaliana*. (**A**) Phenotypes of transgenic *Arabidopsis* lines (S1, S3, S4), WT and UL (unload line) under salt stress and normal conditions. The scale represents 1 cm. (**B**) Survival rate of WT, UL and transgenic lines (S1, S3, S4) seedlings under control (CK) and salt stress. The data represent the mean and standard error of three repetitions. The asterisk above the column indicates that there is a significant difference compared with WT (** *p* ≤ 0.01).

**Figure 5 ijms-23-08007-f005:**
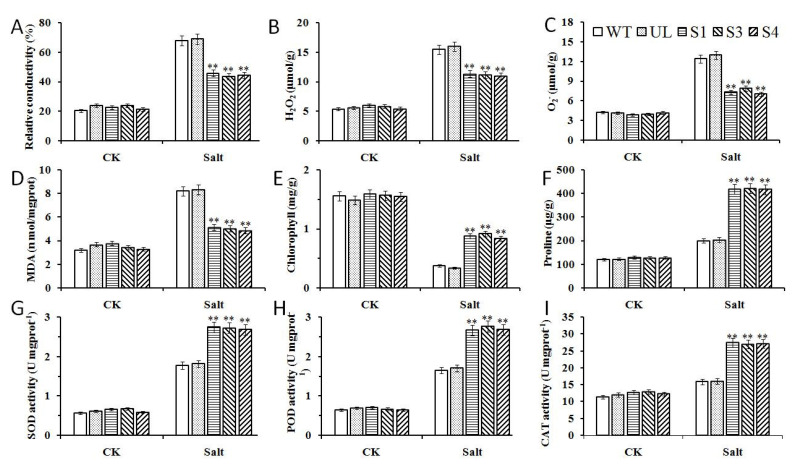
Overexpression of *MxbHLH18* in *A. thaliana* improved salt tolerance. Effects of *MxbHLH18* gene on (**A**) Relative conductivity, (**B**) H_2_O_2_ content, (**C**) O^2−^ content, (**D**) MDA, (**E**) chlorophyll, (**F**) proline content and (**G**) SOD, (**H**) POD, (**I**) CAT enzyme activities in *A. thaliana* under salt stress. Data represent means and standard errors of three replicates. Asterisks above columns indicate significant difference compared to that in WT (** *p* ≤ 0.01).

**Figure 6 ijms-23-08007-f006:**
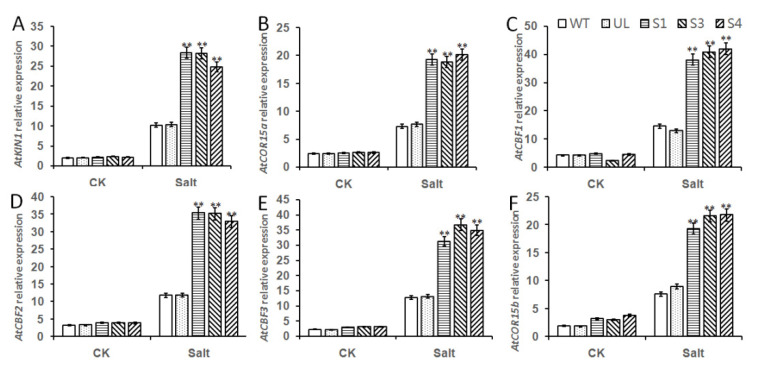
Relative expression levels of salt stress related genes in WT, UL and transgenic *Arabidopsis*. The relative expression levels of *AtK1N1* (**A**), *AtCOR15a* (**B**), *AtCBF1* (**C**), *AtCBF2* (**D**), *AtCBF3* (**E**) and *AtCOR15b* (**F**). Data represent means and standard errors of three replicates. Asterisks above columns indicate significant difference compared to that in WT (** *p* ≤ 0.01).

**Figure 7 ijms-23-08007-f007:**
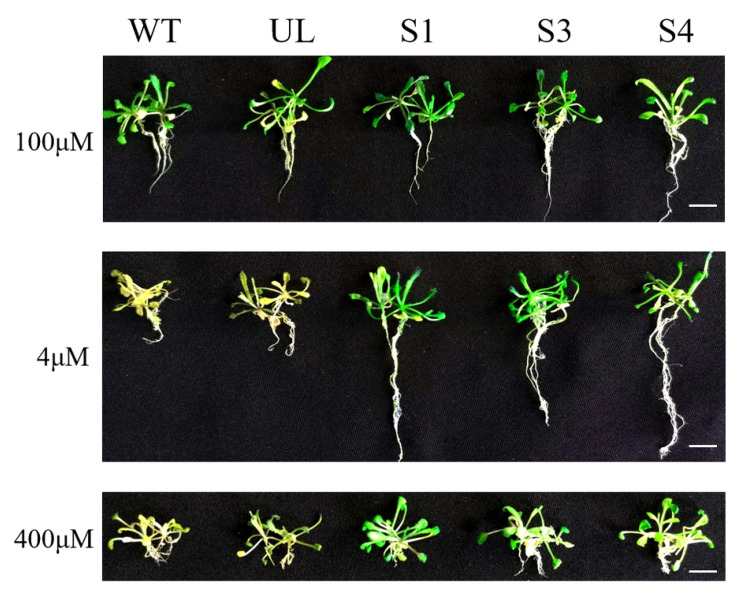
Overexpression of *MxbHLH18* in *A. thaliana* improved Fe stress tolerance. Phenotypes of *MxbHLH18* transgenic *A. thaliana* lines (S1, S3, S4), WT and UL under on normal Fe level (100 μM), low Fe level (4 μM) and high Fe level (400 μM). Scale bar corresponds to 1 cm.

**Figure 8 ijms-23-08007-f008:**
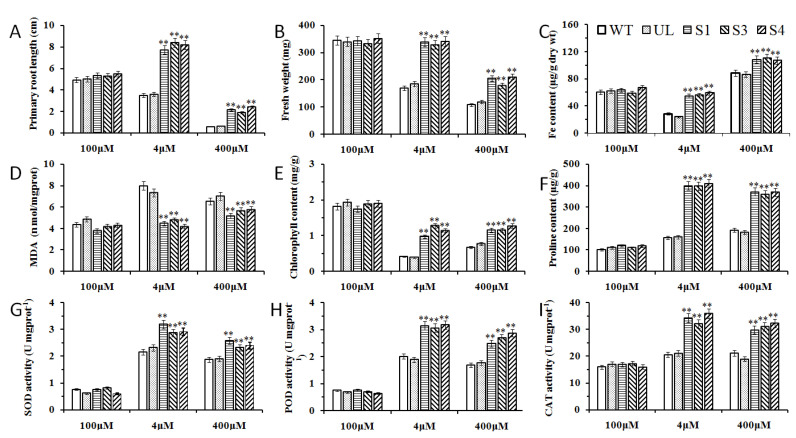
Overexpression of *MxbHLH18* in *A. thaliana* improved Fe tolerance. Effects of *MxbHLH18* gene on (**A**) primary root length, (**B**) fresh weight, (**C**) Fe content, (**D**) MDA, (**E**) chlorophyll, (**F**) proline content and (**G**) SOD, (**H**) POD, (**I**) CAT enzyme activities in *A. thaliana* under Fe stress. The data correspond to the mean and standard error of three repetitions. The asterisk above the column represents significant difference compared with WT (** *p* ≤ 0.01).

**Figure 9 ijms-23-08007-f009:**
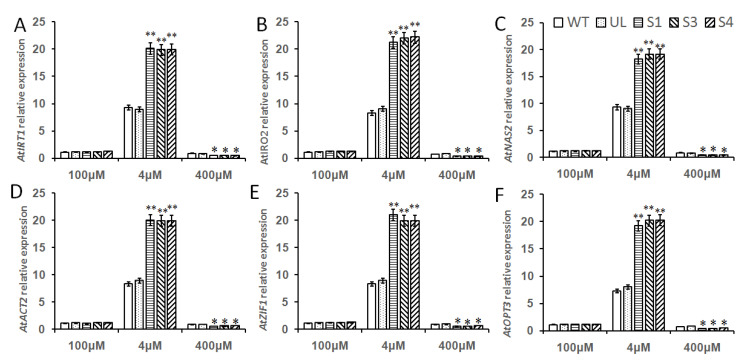
Relative expression levels of iron stress related genes in WT, UL and transgenic *Arabidopsis*. The relative expression levels of *AtIRT1* (**A**), *AtFRO2* (**B**), *AtNAS2* (**C**), *AtACT2* (**D**), *AtZIF1* (**E**) and *AtOPT3* (**F**). Data represent means and standard errors of three replicates. Asterisks above columns indicate significant difference compared to that in WT (* *p* ≤ 0.05, ** *p* ≤ 0.01).

**Figure 10 ijms-23-08007-f010:**
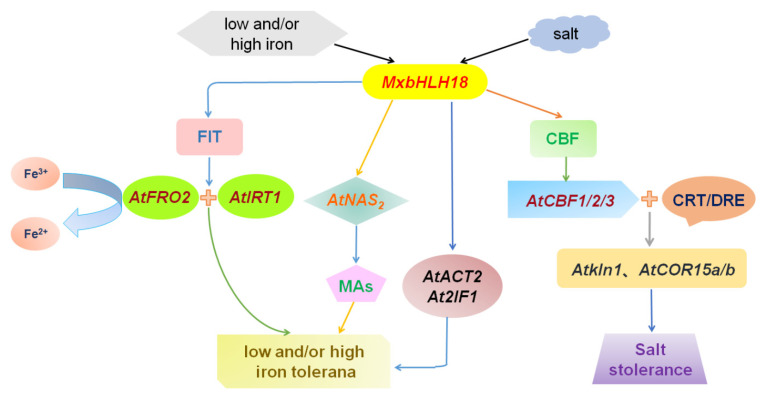
A possible model of *MxbHLH18* response to high salt, low iron and high iron stress. Salt stress induced the expression of *MxbHLH18*, which combined with CBF promoter. Thus, it can promote the expression of *CBF1*, *CBF2* and *CBF3* and directly activate their expression by combining with the CRT/DRE cis-acting elements of the downstream salt stress response genes *AtKIN1*, *AtCOR15a* and *AtCOR15b* which participate in the process of ABA synthesis and ABA signal transduction, so as to regulate the response of plants to salt stress. Due to the stimulation of low iron, *MxbHLH18* gene is overexpressed, bHLH protein and FIT protein form heterodimer, which induces the up-regulated expression of *AtFRO2* and *AtIRT1*. *AtFRO2* promotes the reduction of Fe^3+^ adsorbed on the root epidermis to Fe^2+^. At the same time, the expression of *AtNAS2* was up-regulated, which promoted the synthesis of MAs. In addition, the expression of other key iron stress response genes *AtACT2*, *AtZIF1* and *AtOPT3* located downstream of *MxbHLH18* was also up-regulated. When the signal of high iron stress is felt, the expression of these downstream genes is inhibited, which reduces the absorption and transportation of Fe^2+^ by plant cells, so as to maintain the balance of Fe^2+^ in cells, thereby improving the tolerance of plants to iron stress.

## Data Availability

The original data to this present study are available from the corresponding authors.

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
