# Peer review of "Overexpression of MxbHLH18 Increased Iron and High Salinity Stress Tolerance in Arabidopsis thaliana"

_ijms, 2022, doi:10.3390/ijms23148007_

Round 1

Reviewer 1 Report

The manuscript presented by Liang et al. is focused on the role of a new BHLH transcription factor (TF) gene named xbHLH18 from Malus xiaojinensis in the resistance of Arabidopsis thaliana to salt, high iron and low iron stresses. The authors show that MxbHLH18 is localized in the nucleus and that salt stress and iron stress induces expression of MxbHLH18 in Malus xiaojinensis seedlings. By measuring a series of morphological and physiological indexes such as root length, fresh weight and iron content, contents of chlorophyll, proline, MDA, H2O2 and O2-, activities of SOD, POD and CAT, and the relative conductivity, the authors further show that overexpression of MxbHLH18 enhances the resistance of Arabidopsis to salt, high iron and low iron stresses.

There are some comments regarding this manuscript following below:

Major points:

From the Introduction part of this manuscript, it is clear that many previous studies including the papers published by the same lab of this manuscript have shown that many BHLH TFs play important roles in the resisitance of plants to a series of biotic and abiotic stresses including salt and iron stresses. However, the underlining mechanisms by which these TFs enhance plant resistence to the stresses are not fuuly understood. Although this manuscript adds a new TF gene named XbHLH18 in the resistance of plants to salt and iron stresses, the underlining mechanism(s) by which XbHLH18 enhances plant resistence to stresses is not further explored or discussed in this manuscript.

Minor modification:

1. In line 122, the word “was” was repeated two timens.

2. In line 132, I suggest to add “Under” before the word “current”.

3. In the legend of Figure 4A: (1) the describtion “the asterisk above the column represents a significant difference compared with the expression in the root” in line 145-146 is not consistent with the Figure 4A; (2). (B) should be (B and C) in line 147.

4. In line 153, the describtion “The results of qPCR verification of transgenic Arabidopsis can be seen from Figure 5A.” is not consistent with Figure 5A, which is a phenotypic figure. I suggest that the results of qPCR verification of the overexpression lines of MxbHLH18 should be supplemented in the Supplementary Materials part.

5. “UL” treatment should be explained when it first appears in the figure legend of Figure 5 and text in line 154. 

6. In line 190, there should have a space between number and unit. The same in the text of other places.

7. In lines 231 and 235, “Liu showed” and “Mao analyzed” shoud be corrected to “Liu et al. Showed” and “Mao et al. Analyzed” respectively, because these references have more than one authors. The same in the text of other places.

8. In line 233-234, please check whether the word “resistant” in the sentence “In contrast, 232 bHLH122 loss-of-function mutants were more resistant than WT plants” was used correctly.

9. In lines 232-236, the sentence “More sensitive to NaCl and osmotic stress.” as well as in lines 236-237, the sentence “May be associated with drought and salt stress.” are not complete sentences, and difficuld to be understood.

10. In lines 240-241, I suggest to correct “the transgene stress resistance of plants”.

11. In lines 267-269, the authors stated that “Compared with mature leaves, the expression of MxbHLH18 gene was higher in new leaves and roots of M. xiaojinensis (Figure 4). From this expression pattern, it can be concluded that MxbHLH18 may play an important regulatory role mainly in organs related to stress signals”. I do not understand how the authors only based on this expression pattern to conclude that MxbHLH18 may play an important regulatory role mainly in organs related to stress signals.  

12. In line 318, please remove the unused space In “Trans script”.

13. Before “Author Contributions” section, the “Supplementary Materials” part should be introduced. 

Author Response

Reviewer 1

From the Introduction part of this manuscript, it is clear that many previous studies including the papers published by the same lab of this manuscript have shown that many BHLH TFs play important roles in the resisitance of plants to a series of biotic and abiotic stresses including salt and iron stresses. However, the underlining mechanisms by which these TFs enhance plant resistence to the stresses are not fuuly understood. Although this manuscript adds a new TF gene named MXbHLH18 in the resistance of plants to salt and iron stresses, the underlining mechanism(s) by which MXbHLH18 enhances plant resistence to stresses is not further explored or discussed in this manuscript.

Response: We appreciate your valuable suggestions and advices on our manuscript. I think they are very helpful and important, and revisions had been made in the revised manuscript accordingly.

Here I would like to response the comments and add some explanations as follows.

At the same time, the content about MxBHLH18 regulating downstream stress-related genes was supplemented. Supplementary materials and pictures have also been corrected.

The language of manuscript have also been revised.

The modified part is marked red.

Minor modification:

  1. In line 122, the word “was” was repeated two timens.

Response: Yes, we have accepted your suggestion and corrected this part.

  1. In line 132, I suggest to add “Under” before the word “current”.

Response: Yes, we have accepted your suggestion and added“Under” before the word “current”.

  1. In the legend of Figure 4A: (1) the describtion “the asterisk above the column represents a significant difference compared with the expression in the root” in line 145-146 is not consistent with the Figure 4A; (2). (B) should be (B and C) in line 147.

    Response: Yes, we have accepted your suggestions and changed these parts.

  1. In line 153, the describtion “The results of qPCR verification of transgenic Arabidopsis can be seen from Figure 5A.” is not consistent with Figure 5A, which is a phenotypic figure. I suggest that the results of qPCR verification of the overexpression lines of MxbHLH18 should be supplemented in the Supplementary Materials part.

Response: Yes, we have accepted your suggestions and corrected these contents. Due to my negligence, the article description is wrong, so this part “The results of qPCR verification of transgenic Arabidopsis can be seen from Figure 5A.” has been deleted.

  1. “UL” treatment should be explained when it first appears in the figure legend of Figure 5 and text in line 154.

 Response: Yes, we have accepted your suggestion and added these content.

  1. In line 190, there should have a space between number and unit. The same in the text of other places.

Response: Yes, we have accepted your suggestion and revised these part.

  1. In lines 231 and 235, “Liu showed” and “Mao analyzed” shoud be corrected to “Liu et al. Showed” and “Mao et al. Analyzed” respectively, because these references have more than one authors. The same in the text of other places.

Response: Yes, we have accepted your suggestion and corrected these content.

  1. In line 233-234, please check whether the word “resistant” in the sentence “In contrast, 232 bHLH122 loss-of-function mutants were more resistant than WT plants” was used correctly.

Response: Yes, we have accepted your suggestion and corrected this content.

  1. In lines 232-236, the sentence “More sensitive to NaCl and osmotic stress.” as well as in lines 236-237, the sentence “May be associated with drought and salt stress.” are not complete sentences, and difficuld to be understood.

 Response: Yes, we have accepted your suggestion and added these content.

  1. In lines 240-241, I suggest to correct “the transgene stress resistance of plants”.

Response: Yes, we have accepted your suggestion and corrected this content.

  1. In lines 267-269, the authors stated that “Compared with mature leaves, the expression of MxbHLH18 gene was higher in new leaves and roots of M. xiaojinensis (Figure 4). From this expression pattern, it can be concluded that MxbHLH18 may play an important regulatory role mainly in organs related to stress signals”. I do not understand how the authors only based on this expression pattern to conclude that MxbHLH18 may play an important regulatory role mainly in organs related to stress signals.

Response: Yes, we have accepted your suggestion, corrected this content and made a more accurate description.

  1. In line 318, please remove the unused space In “Trans script”.

Response: Yes, we have accepted your suggestion and removed this unused space.

  1. Before “Author Contributions” section, the “Supplementary Materials” part should be introduced.

Response: Yes, we have accepted your suggestion and added these content.

Reviewer 2 Report

It is an interesting and topical study with results that can be transferred to the biotechnology industry for plant improvement. The study is very well written, clear, and the conclusions are supported by the results obtained, these being the main strong points. I have only a few specific recommendations:

L38 specifies what TF is.

L151 please pay attention to the value of P, you should use * p ≤ 0.05, ** p ≤ 0.01, the same is true for L166, L187 and L217.

I think it would be useful to specify in each title of figure the species to which it refers.

L154 tables S2, S3 and S4 are missing from the supplementary material, please enter them. Also, I do not think that Table S1 is correctly quoted here and S2 is not quoted anywhere in the text, please be careful and correct.

Author Response

It is an interesting and topical study with results that can be transferred to the biotechnology industry for plant improvement. The study is very well written, clear, and the conclusions are supported by the results obtained, these being the main strong points. I have only a few specific recommendations:

Response: We appreciate your valuable suggestions and advices on our manuscript. I think they are very helpful and important, and revisions had been made in the revised manuscript accordingly.

Here I would like to response the comments and add some explanations as follows.

At the same time, the content about MxBHLH18 regulating downstream stress-related genes was supplemented. Supplementary materials and pictures have also been corrected.

The language of manuscript have also been revised.

The modified part is marked red.

  1. L38 specifies what TF is.

   Response: Yes, we have accepted your suggestion and corrected this part.

  1. L151 please pay attention to the value of P, you should use * p ≤ 0.05, ** p ≤ 0.01, the same is true for L166, L187 and L217.

Response: Yes, we have accepted your suggestion and corrected this part.

  1. I think it would be useful to specify in each title of figure the species to which it refers.

Response: Yes, we have accepted your suggestion and corrected these parts.

  1. L154 tables S2, S3 and S4 are missing from the supplementary material, please enter them. Also, I do not think that Table S1 is correctly quoted here and S2 is not quoted anywhere in the text, please be careful and correct.

Response: Yes, we have accepted your suggestion and corrected these parts. Tables2, S3, S4 were not involved in the text. Due to our negligence, the transgenic lines ''S1, S3, S4,'' wre tagged "S2, S3, S4", which has been corrected, including words and pictures.

Round 2

Reviewer 1 Report

The authors have sufficiently improved this manuscript according to the reviewer`s comments. However, I think there are several points that should be improved.

1. The main results and conclusion about MxBHLH18 regulating downstream stress-related genes should be included in the Abstract part.

2. In lines 249-254 and 357-359, the results of Figure 9 under high iron stress were not clearly or correctly presented. Figure 9 clearly showed that, compared with the control (100 μM), the expression levels of the 6 genes checked were all up-regulated in all Arabidopsis lines (WT, UL, S1, S3 and S4) under high iron stress, but there were no significant difference between MxBHLH18 transgenic lines (S1, S3 and S4) and WT or UL, which is different from the results that the up-regulation of the 6 genes in MxBHLH18 transgenic lines was significantly higher than that in WT and UL under low iron stress. Thus, I do not understand why the authors stated that “after high iron stress treatment, the expression of these 6 genes was not significantly up-regulated, and the up-regulated expression was not significantly different, indicating that these 6 genes were not regulated under high iron stress”.

3. Based on the results of Figure 9, it can be concluded that MxBHLH18-enhanced plant resistance to high iron stress was not dependent on the expression level of AtIRT1, AtFRO2, AtNAS2, AtACT2, AtZIF1 and AtOPT3, which suggests that the mechanism(s) by which XbHLH18 enhances plant resistance to high iron stress is different from that to low iron stress. However, this important difference was not clearly stated in the Results and Discussion parts and was also not reflected in the model of Figure 10.

4. In line 222, “Figure 6” should be corrected to “Figure 7”.

5. In line 278, the word “more” was repeated two times.

6. In line 301, “Zhou found” should be “Zhou et al. found”.

Author Response

The authors have sufficiently improved this manuscript according to the reviewer`s comments. However, I think there are several points that should be improved.

Response: We appreciate your valuable suggestions and advices on our manuscript. I think they are very helpful and important, and revisions had been made in the revised manuscript accordingly. Here I would like to response the comments and add some explanations as follows. The language of manuscript have also been revised. The modified part is marked red.

1. The main results and conclusion about MxBHLH18 regulating downstream stress-related genes should be included in the Abstract part.

Response: Yes, we have accepted your suggestion and added these important parts.

2. In lines 249-254 and 357-359, the results of Figure 9 under high iron stress were not clearly or correctly presented. Figure 9 clearly showed that, compared with the control (100 μM), the expression levels of the 6 genes checked were all up-regulated in all Arabidopsis lines (WT, UL, S1, S3 and S4) under high iron stress, but there were no significant difference between MxBHLH18 transgenic lines (S1, S3 and S4) and WT or UL, which is different from the results that the up-regulation of the 6 genes in MxBHLH18 transgenic lines was significantly higher than that in WT and UL under low iron stress. Thus, I do not understand
why the authors stated that after high iron stress treatment, the expression of these 6 genes was not significantly up-regulated, and the up-regulated expression was not significantly different, indicating that these 6 genes were not regulated under high iron stress.

Response: Yes, we have accepted your suggestion and corrected the test data again. The test results are also re-reflected in Figure 9, and the test results are described more accurately.

3. Based on the results of Figure 9, it can be concluded that MxBHLH18-enhanced plant resistance to high iron stress was not dependent on the expression level of AtIRT1, AtFRO2, AtNAS2, AtACT2, AtZIF1 and AtOPT3, which suggests that the mechanism(s) by which XbHLH18 enhances plant resistance to high iron stress is different from that to low iron stress.
However, this important difference was not clearly stated in the Results and Discussion parts and was also not reflected in the model of Figure 10.

Response: Yes, we have accepted your suggestion and corrected the problem as well as added some explanations as follows:
First, we have described the experimental results more accurately. MxBHLH18 can promote the up-regulated expression of downstream genes under low iron stress and inhibit the expression of downstream genes under high iron stress. Therefore, MxBHLH18 can regulate the expression of these six downstream genes and improve the ability of plants to resist iron stress. This is consistent with the research results of Gang Liang, Honghong Bai, Yan Cui et al. and Lili Yin et al., and this important part
has also been explained more clearly in the summary, results and discussion. Please refer to this article for details, and then we will do further research in subsequent experiments.

4. In line 222, Figure 6 should be corrected to Figure 7.

Response: Yes, we have accepted your suggestion and corrected this parts.

5. In line 278, the word more was repeated two times.

Response: Yes, we have accepted your suggestion and revised it.

6. In line 301, Zhou found should be Zhou et al. found.

Response: Yes, we have accepted your suggestion and added this part